# The key role of atmospheric absorption in the Asian Summer Monsoon response to dust emissions in CMIP6 models

Alcide Zhao[1,2], Laura J. Wilcox[1,2], Claire L. Ryder[1]

[1]Department of Meteorology, University of Reading, Reading, UK
[2]National Centre for Atmospheric Science, UK

*Correspondence to*: Claire L. Ryder (c.l.ryder@reading.ac.uk)

**Abstract.**

We investigate the Asian Summer Monsoon (ASM) response to global dust emissions in the Coupled Model Intercomparison
Project Phase 6 (CMIP6) models, which is the first CMIP to include an experiment with a doubling of global dust emissions
relative to their preindustrial levels. Thus, for the first time, the inbuilt influence of dust on climate across a range of climate
models being used to evaluate and predict Earth's climate can be quantified. We find that dust emissions cause a strong
atmospheric heating over Asia that leads to a pronounced energy imbalance. This results in an enhanced Indian Sumer
Monsoon (ISM) and a southward shift of the Western Pacific Intertropical Convergence Zone (ITCZ), consistent across
models, with the strength of the ISM enhancement increasing with the magnitude of atmospheric dust shortwave absorption,
driven by dust optical depth changes. However, the East Asian Summer Monsoon response shows large uncertainties across
models, arising from the diversity in models' simulated dust emissions, and in the dynamical response to these changes. Our
results demonstrate the central role of dust absorption in influencing the ASM, and the importance of accurate dust simulations
for constraining the ASM and the ITCZ in climate models.

**1 Introduction**

Mineral dust is the most abundant aerosol type by mass in the Earth's atmosphere (Kok et al., 2018; Gliss et al., 2021), and
their emissions have at least doubled since preindustrial times (Hooper and Marx, 2018). Dust aerosols play an important role
in the Earth's radiation balance and climate system by interacting with radiation, clouds, and ecosystems during its life cycle
(Carslaw et al., 2010; Mahowald et al., 2010; Kok et al., 2018; Chaibou et al., 2020; Jin et al., 2021). Overall, dust serves as a
cooling agent at the top of the atmosphere (TOA) over all but the brightest surfaces (Chaibou et al., 2020). Dust also heats the
atmospheric column by absorbing solar radiation but also cools the atmosphere through terrestrial radiation interactions,
thereby perturbing the vertical temperature profile (Balkanski et al., 2021; Ryder, 2021). However, our knowledge of dust-
climate interactions, including the magnitude and sign of dust's radiative effect, remains highly uncertain due to incomplete
understanding of its physical and chemical properties (Formenti et al., 2011; Richter and Gill, 2018; Di Biagio et al., 2019;

Adebiyi and Kok, 2020), life cycles (Shao et al., 2011; Kok et al., 2021b; Wu et al., 2020), and interactions with other components of the Earth system (Karydis et al., 2017; Chaibou et al., 2020; Li et al., 2021), as well as the challenges of incorporating these processes into models.

Present-day global dust emissions are confined primarily to the Northern Hemisphere tropical and subtropical regions (i.e., the
so-called dust-belt (Shi et al., 2021)), with around 30-40% emitted from the Asian source regions (Kok et al., 2021a). There are numerous studies investigating Asian dust-climate interactions, and particularly their links to the Indian Summer Monsoon (ISM) (Sun et al., 2012; Chen et al., 2017; Wang et al., 2020; Jin et al., 2021). It is found that dust impacts the ISM through many different pathways including the elevated heat pump mechanism (Lau et al., 2006), snow-darkening feedbacks (Sarangi et al., 2020), and dust-cloud interactions (Karydis et al., 2017). However, most of these mechanisms are are subject to large
uncertainties in model physics and parameters (Jin et al., 2021). Unfortunately, these uncertainties are very difficult to constrain using available observations. Compared to the ISM, there are larger uncertainties in our understanding of the interactions between dust and the East Asian climate, including the East Asian Summer Monsoon (EASM) (Sun et al., 2012; Chen et al., 2017; Wang et al., 2020). Dust emissions have the potential to impact both the ISM and the EASM, collectively known as the Asian Summer Monsoon (ASM).


The availability of the Coupled Model Intercomparison Project Phase 6 (CMIP6; Eyring et al. (2016)) experiments offer a great opportunity to understand the climate impacts of dust emissions and the role dust plays in the latest generation of climate models. Zhao et al. (2022) examined the global and regional simulation of dust in 16 CMIP6 models in the Atmospheric Model Intercomparison Project (AMIP) experiments compared to observations and reanalyses, finding that most models captured
features such as spatial distribution and seasonal cycles of dust well, but dust emission and deposition were poorly represented, and that the ranges of simulated dust burden and optical depth across models are larger than that of previous model generations. Several publications have examined dust simulation and response to climate change in other CMIP6 experiments. Aryal and Evans (2021) examined dust sensitivity to drought in historical and future SSP585, showing that soil moisture is a better indicator of dust variability than precipitation, highlighting the importance of the land surface in simulating the dust cycle
accurately. Aryal and Evans (2023) and Zhou et al. (2023) explored the response of dust emissions and surface concentrations to temperature and precipitation/soil moisture changes, finding substantial regional variability. Zhao et al. (2023) found that overall, dust loading increases globally by the end of the twenty-first century in CMIP6 model simulations, though this is dependent on the future scenario and region, with East Asia and the western Pacific showing decreasing dust load due to increasing precipitation in these regions. Contrastingly, Mao et al. (2021) suggest an increase in East Asian dust emissions in
CMIP6 future simulations due to enhanced frequency of circulation patterns connected to extreme dust events. Li and Wang (2022) explored drought-dust relationships over the southeastern USA in CMIP6 historical simulations. Gomez et al. (2023)

highlighted the important role of interactions between dust and the West African monsoon in contributing to future air quality degradations.

The CMIP6 Aerosols and Chemistry Model Intercomparison Project (AerChemMIP; Collins et al. (2017)) has for the first time included a doubled-dust experiment alongside single forcing experiments with other aerosol species. This allows us to consistently isolate and quantify the impacts of dust emissions in multiple state-of-the-art climate models. Although dust aerosols have been included in previous CMIP experiments as well as the latest CMIP6 historical, AMIP and future SSP experiments, these experiments do not allow the isolation of the specific effect of dust on radiation and climate in a multi-
model context. It is important to understand the role and extent of dust in impacting climate in the CMIP6 simulations, where the effects of dust on climate (through mechanisms such as surface, atmospheric or top of atmosphere radiative effects and subsequent complex impacts on atmospheric circulation) are present, but not explicit. For the first time, the new AerChemMIP experiments allow this to be diagnosed.

We present a multi-model study to determine the atmospheric response to a change in global dust emissions in Asia based on two sets of the AerChemMIP simulations from seven CMIP6 models (Section 2). Dust radiative forcing, temperature, and precipitation responses, as well as the circulation changes and mechanisms are presented in Section 3. Our major findings and their implications are summarised in Section 4.

## 2 Models and Simulations

To explore the impact of dust emissions, we used two sets of time-slice simulations from seven participating CMIP6 models shown in Table 1, which provided dust diagnostics. We include all 7 models regardless of how well (or poorly) they represent the dust cycle (Zhao et al., 2022), in order to firstly understand the implicit effect of dust in climate simulations in general within CMIP6 models, and secondly to avoid limiting further the number of models analysed. Even if models do not simulate the dust cycle well, it is important to understand how dust may be influencing the climate and circulation in CMIP6 models.
The base simulation (piClim-control) has all forcings fixed at preindustrial (year 1850) levels. The AerChemMIP perturbation simulation (piClim-2xdust) is identical to piClim-control except that dust emissions are doubled globally. The CMIP6 models reproduce major features of global dust processes well (Zhao et al., 2022), including the spatial patterns of global dust emissions and dust aerosol optical depth (DOD). Dust emissions were calculated online in all the seven models in piClim-control, and were doubled in piClim-2xdust by scaling the dust emission parameterisations (Collins et al., 2017). As such, we
define the climate impacts of dust emissions as the difference between piClim-2xdust and piClim-control (i.e., piClim-2xdust minus piClim-control). Sea surface temperatures and sea ice distributions were prescribed as 1850 climatological averages in both simulations. Therefore, the responses presented here represent the fast response of the climate system due to rapid

adjustments of the atmosphere to changes in energy balance as a direct result of dust emissions (Ganguly et al., 2012; Samset et al., 2016; Zanis et al., 2020).


Table 1 also includes pertinent information relating to the dust scheme in each model, including references for the wind-driven dust emission scheme applied. Table 1 includes the type of size distribution utilized (i.e. sectional or modal) and its diameter range or modal values. We note that the largest size represented is 63 um by UKESM1-0-LL. Maximum size represented by modal schemes is difficult to assess, though it is likely that these schemes represent super-coarse dust particles poorly (e.g.

Jones et al. (2022)) In most models dust does not act as a CCN. However, in some models the role of dust is not isolated from other aerosols, as a single mode may comprise a mixture of aerosol species, including dust, the combination of which can act as CCN but is not driven by dust, and thus the role of dust through this pathway is expected to be very small (e.g. MPI-ESM-1-2-HAM). Two models (NorESM2-LM and MPI-ESM-1-2-HAM) include dust acting as ice nuclei (IN) . All models include the interaction between dust and LW radiation (in addition to SW radiative interactions) which occurs due to the larger size of

dust relative to other aerosol species.

Table 1 also includes information on the complex refractive index (CRI) used for dust in each model, of which the real and imaginary parts determine the scattering and absorption properties of dust respectively. The imaginary refractive index (IRI) of dust in models has received attention recently due to the publication of updated laboratory IRI data (Di Biagio et al., 2019),

with several studies demonstrating that current climate models overestimate the amount of absorption due to dust ((Adebiyi et al., 2023b). Values for the models and simulations shown in Table 1 encompass IRI values ranging from $1.1 \times 10^{-3}$ to $8 \times 10^{-3}$ at mid-visible wavelengths, with all models except CNRM-ESM2-1 using values smaller than $2.4 \times 10^{-3}$. In this study the mean and median model IRI are $2.56 \times 10^{-3}$ and $1.47 \times 10^{-3}$ respectively. All models except CNRM-ESM2-1 fall within the range suggested by the laboratory data of Di Biagio et al. (2019), while CNRM-ESM2-1 lies on the upper edge of the range indicated

by measurements. It is notable that these CMIP6 models have a lower median IRI than those evaluated by Adebiyi et al. (2023b) found to overestimate absorption, and only one lies in the range of suggested over-absorption suggested by Wang et al. (2024) ; i.e. most of the CMIP6 models investigated here simulate plausible IRI for dust in the mid-visible spectral region.

The relative amount of absorption occurring due to dust aerosol is determined by the single scattering albedo (SSA) (Highwood

and Ryder, 2014), which is determined by the CRI applied in a model, dust shape and the modelled and evolving (in space and time) size distribution. Although the SSA is a good indicator of dust absorption in the atmosphere, we do not report it here since it will also vary in space and time due to its dependence on the size distribution. Simulated dust mass data as a function of size is not available for these CMIP6 experiments. In models utilizing modal dust schemes, optical properties are typically calculated for a mixture of aerosol species existing within each mode, so reporting SSA is not meaningful for dust specifically.

Finally, only a few models specifically document dust SSA (often due to its spatially variable nature) with the CRI being a much more commonly documented variable (Table 1).

| Model | Variant label | Resolution (lon × lat × Lev) | Model years | Dust size representation and boundaries (µm) | Dust as CCN/IN | Global JJA mean effective radiative forcing (W m$^{-2}$) | References | Dust Refractive Index at 550nm[5] | LW Dust interactions? | Dust Emission Scheme |
|---|---|---|---|---|---|---|---|---|---|---|
| CNRM-ESM2-1 | r1i1p1f2 | 1.4° × 1.4°× 91L | 30 | Sectional; 3 bins (0.01, 1.0, 2.5, 20) | N/N | 0.08 | Séférian et al. (2019); Michou et al. (2015) | 1.51-0.008i | Y | Marticorena and Bergametti (1995); Kok (2011); Nabat et al. (2012); Nabat et al. (2015) |
| GFDL-ESM4 | r1i1p1f1 | 1.25° × 1°× 49L | 30 | Sectional; 5 bins (0.2, 2, 4, 6, 20) | N[1]/N | -0.07 | Dunne et al. (2020); (Horowitz et al., 2020; Donner et al., 2011) | 1.52-1.47i (SW from Balkanski et al. (2007) LW from Volz (1973) | Y | Ginoux et al. (2001); Evans et al. (2016) |
| GISS-E2-1-G | r1i1p3f1 | 2.5° × 2°× 40L | 41 | Sectional; 6 bins (0.2, 2, 4, 8, 16, 32) | Y[4]/N | -0.11 | Kelley et al. (2020); Bauer et al. (2020); OMA scheme | 1.56-0.002i based on Sinyuk et al. (2003) in SW; Volz (1973) for λ>2µm | Y | Miller et al. (2006); Cakmur et al. (2006) |
| IPSL-CM6A-LR-INCA | r1i1p1f1 | 1.25° × 1.27°× 79L | 30 | Modal: 1 lognormal mode, MMD (GSD): 2.5 (2) | N/N Dust treated as insoluble | -0.19 | Boucher et al. (2020); Hourdin et al. (2020) INCA Hauglustaine et al. (2014) | 1.52-0.00147i (Balkanski et al. 2007)[3] | Y | Schulz et al. (2007) |
| MPI-ESM-1-2-HAM | r1i1p1f1 | 1.875° × 1.875° × 47L | 40 | Modal: 2 modes with median particle diameter boundaries (GSD): 0.01-0.1 (1.59); >0.1 (2.0) | N[2]/Y | -0.13 | Mauritsen et al. (2019); Neubauer et al. (2019); Tegen et al. (2019) | 1.52+0.0011i (Kinne 2013) | Y | Marticorena and Bergametti (1995); Tegen et al. (2002); Cheng et al. (2008); Heinold et al. (2016) |

| NorESM 2-LM | r1i1 p1f1 | 2.5° × 1.875° × 32L | 30 | Modal: 2 modes: accumulation and coarse;NMD (GSD) 0.44 (1.59), 1.26 (2.0) | Y/Y | 0.04 | Seland et al. (2020); Kirkevåg et al. (2018) | 1.53-0.0024i | Y | Zender et al. (2003) (DEAD model) |
|---|---|---|---|---|---|---|---|---|---|---|
| UKESM 1-0-LL | r1i1 p1f4 | 1.875° × 1.25°× 85L | 45 | 6 bins :0.064-0.2, 0.63, 2.0, 6.32, 20, 63 | N/N | 0.13 | Bellouin et al. (2011); Mulcahy et al. (2020); Woodward et al. (2022) | 1.52-0.00147i Balkanski et al. (2007) | Y; includes LW dust scattering | Marticorena and Bergametti (1995) |

Table 1: Details of CMIP6 models used in this study. Bin sizes give limits of each size bin. MMD indicates Mass Median Diameter, GSD indicates Geometric Standard Deviation; NMD number median diameter. [1] Does not act as a CCN, however in low-sufate regions dust impacts the sulfate mass distribution. [2] Dust as part of a mixed species mode may act as CCN, though the role of dust would be small. [3] Personal communication, Dr Claudia di Biagio. [4] CCN calculated from total aerosol mass [5] further spectral information given where available.

For each model and experiment, a simulation of at least 30 years is available. All model data are interpolated to a 2x2-degree
horizontal grid when computing the multi-model mean (MMM). We used a first order conservative interpolation for fields that
request integral of the source field across the regridding – dust emissions, for example. For all other variables, we used bilinear
interpolation. We focus on the response over Asia (box in Figure S1f) in the summertime (June-July-August; JJA) when the
Asian Summer Monsoon (ASM) is fully established (Li et al., 2016; Zhao et al., 2019; Jin et al., 2021). All the changes
presented here are due to a doubling of global dust emissions, and we refer to this as 'dust emissions' for simplicity.


We diagnosed the dust effective radiative forcing (ERF) as the differences in net radiative fluxes between piClim-2xdust and
piClim-control at the top-of-the-atmosphere (TOA) and at the surface (Forster et al., 2016). We then defined change in
atmospheric absorption due to dust emissions as the difference between TOA and surface ERF. The response to dust emissions
such as changes in surface temperature and precipitation are calculated as averages of JJA means of the last 30 years of each
simulation (Table 1). We tested the statistical significance of the response at the $p \leq 0.1$ confidence level using the Monte-
Carlo test. We have also identified regions where there are inconsistent responses across models, defined as regions where $\leq 4$
of the 7 models have the same sign as the MMM. Radiative fluxes are given as total (shortwave plus longwave) unless
specifically stated otherwise. Clear-sky ERFs are obtained by a double-call to the radiation scheme within a model and
represent an atmosphere without clouds (e.g. Ghan (2013)). Cloud ERFs are calculated as the difference between all-sky and
clear-sky ERFs.

**3 Summertime climate responses to dust emissions**

The models' simulated changes in dust emissions and DOD are shown in Figure 1, while the JJA climatologies of DOD,
precipitation, 850-hPa winds and sea level pressure fields are included in Supplement Figures S1-S3 for reference. We note
the large diversities in models' simulated dust climatology (Figure 1, S1), and hence in the changes to DOD (Figure S4)
because of doubling global dust emissions. The diversity in DOD climatology (Figure 1c, d) also results in statistically
insignificant changes in DOD due to doubled dust emissions in the dustiest regions (hatches in Figure 1f), despite DOD
magnitudes broadly doubling. Such inter-model diversities are most pronounced over the Chinese deserts and East Asia where
a half of the models simulate very little dust emission. The models also simulate very different monsoon climatologies (Figure
S2, S3), which will contribute to differences in the DOD distribution through dust transport and wet deposition differences,
and likely to differences in the response of the monsoon to the dust forcing (Liu et al., 2018).

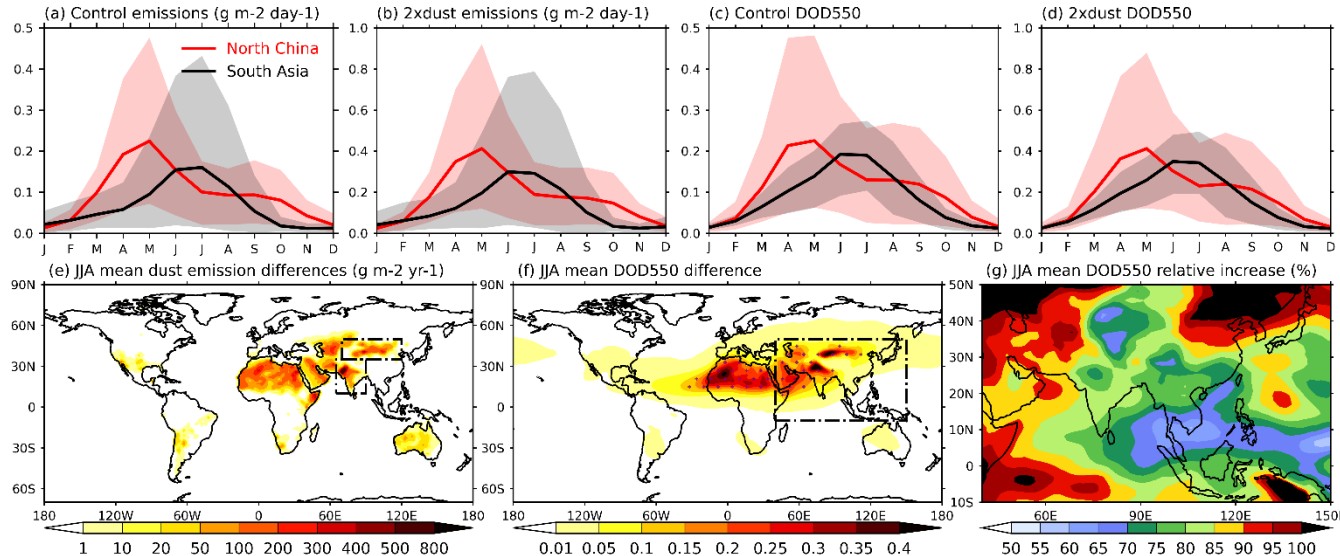

**Figure 1: Model simulated (30-year mean) seasonal cycles of (a, b) dust emissions (g m⁻² day⁻¹) and (c, d) DOD at 550 nm (DOD550) over the Chinese desert (red) and South Asia (black; see boxes in (e)) in (a, c) piClim-control and (b, d) piClim-2xdust. Shadings represent the 5ᵗʰ-95ᵗʰ percentiles of the multi-model ensemble spread. Maps show multi-model mean of JJA mean differences in (e) dust emissions (g m⁻² yr⁻¹), (f) DOD550, (g) the multi-model mean of percentage increase in DOD550 relative to the piClim-Control climatology over Asia (denoted by box in (f)). Purple hatches in (f) denote statistical insignificance at the 10% level of the multi-model mean DOD changes.**

### 3.1 Changes in radiative forcing and clouds

Figure 2 shows the spatial patterns, and zonal mean profiles, of the ERF over Asia due to dust emissions. Those for individual models can be found in the supplement. The clear-sky ERF at TOA in the MMM shows a general negative forcing (Figure 2a) due to the direct dust-radiation interactions (i.e., scattering and absorption), resulting from a mostly negative TOA clear-sky SW ERF (Figure S6) contrasted with a smaller positive LW TOA clear-sky ERF (Figure S7). There is however a positive clear-sky TOA ERF pattern that is confined over the bright surface of the Arabian Peninsula, as well as over South and Southeast Asia (particularly in CNRM-ESM2-1, GFDL-ESM4 and UKESM1-0-LL; Figure S5). The spatial pattern of all-sky TOA ERF in the MMM (Figure 2b) resembles that of clear-sky over the land, yet large differences exist across models (Figure S8). Over the Indian subcontinent there is more inter-model agreement in the all-sky TOA ERF (Figure 2a) than the clear-sky (Figure 2b). This is due to a large uncertainty in the sign of models' individual clear-sky TOA ERF (Figure S5) resulting from how the magnitudes of the mostly negative SW clear-sky TOA ERF (Figure S6) and positive LW clear-sky TOA ERF (Figure S7) cancel out, producing varying signs of the total TOA dust clear-sky ERF (Figure S5). Additionally, the net warming effect of clouds in this region shirts the all-sky TOA ERF to positive values across most models (Figure S8), resulting in the better inter-model agreement seen in Figure 2b. Over South Asia, all models but IPSL-CM6A-LR-INCA (-1.73 W m⁻²) simulate a positive all-sky TOA ERF (Figure S8, 0.01-3.38 W m⁻²). Comparing the clear-sky atmospheric absorption (Figure S18) to the

all-sky version (Figure 1c; S21) reveals that most of the increased heating from the Arabian peninsula, across the Indian ocean
185    to Southern India, as well as around the Chinese deserts, is driven by dust-induced atmospheric absorption.

(a) TOA clear-sky net ERF (W m-2)
(b) TOA all-sky net ERF (W m-2)
(c) Atmospheric absorption (W m-2)
(d) Surface all-sky net ERF (W m-2)
(e) Atmospheric absorption (W m-2)
(f) Surface ERF (W m-2)

CNRM-ESM2-1
GFDL-ESM4
GISS-E2-1-G
IPSL-CM6A-LR-INCA
MPI-ESM-1-2-HAM
NorESM2-LM
UKESM1-0-LL
Multi-model mean

**Figure 2: JJA mean changes in radiative fluxes (W m$^{-2}$) due to doubled dust emissions. Maps show multi-model mean differences in (a) clear-sky effective radiative forcing (ERF) at the top-of-the-atmosphere (TOA), (b) TOA all-sky net ERF, (c) all-sky atmospheric total (shortwave plus longwave) absorption, and (d) surface all-sky net ERF. Green hatches denote where ≤4 models have the same sign as the multi-model mean. Curves show the zonal mean differences in (e) net atmospheric absorption and (f) surface ERF averaged between 40°E-150°E. Coloured curves represent individual models and black curves the multi-model mean.**

Dust emissions result in significant all-sky atmospheric heating through dust absorption above land and the Arabian Sea (Figure 2c). This heating is robust across all models (Figure S21), producing a MMM of 1.58 (0.23-2.94) W m$^{-2}$ over Asia (box in Figure 1f), which is dominated by the shortwave radiative heating that is partially cancelled out by the longwave radiative cooling (Figures S11, S21-23). The all-sky atmospheric heating is particularly prominent over South Asia (4.28 (1.01-9.59) W m$^{-2}$). As a result of the dust-induced atmospheric absorption, there is a pronounced negative surface all-sky ERF over land (Figure 2d, S15). Comparing all-sky and clear sky surface ERFs (Figures S12 and S15) reveals that the net surface cooling in these regions is driven by the changes in dust rather than cloud.

Changes in the spatial pattern of total cloud fraction (Figure 3a, S27) over Asia, and especially over southern Asia (0.26%-3.49%), show common patterns across models, generally showing increased cloud fraction in these regions except MPI-ESM-1-2-HAM (-0.37%). These changes come from high cloud increases (above 200 hPa) over the Indian subcontinent (Figure 3c) in all models except MPI-ESM-1-2-HAM,  and broad decreases in cloud fraction over the Pacific Ocean above 400 hPa (Figure 3d).  Changes in high cloud are also observed over the Arabian Sea and East China (not shown). Changes in mid-level clouds (700-200 hPa) above the Chinese deserts (Figure 3b) vary in sign between models. The cloud changes over South Asia and the Pacific Ocean are associated with changes in the large-scale atmospheric circulation (Section 3.3) rather than increases in dust which may modify ice cloud microphysical properties (Figure S31). Interestingly, the two models where dust acts as an IN (MPI-ESM-1-2-HAM and NorESM2-LM) show opposite responses in terms of changes in cloud fraction (Figure 3b and c).

The increased cloud across the Arabian Sea/Southern Asia region results in a weak negative TOA SW cloud ERF (Figure S25), a positive TOA LW cloud ERF (Figure S26), and a positive TOA total cloud ERF (Figure S24). This reduces the magnitude of the negative clear-sky TOA ERF over the Arabian Sea but strengthens the positive values over the Indian subcontinent, resulting in the land-sea contrast in all-sky ERF seen in Figure 2b in this region. This indicates that the all-sky TOA ERF from dust dominates that from cloud over the Arabian Sea, while both dust and cloud act together to generate a positive value over the Indian subcontinent. In terms of the clear-sky atmospheric absorption for this region, dust causes a widespread large atmospheric heating in the SW and cooling in the LW, producing a total heating (Figures S18-20). In addition, increased cloud generates a LW heating effect, which shifts the clear-sky (i.e. dust) LW atmospheric absorption from negative (Figure S20) to positive over ocean (Figure S23) and reduces the magnitude of the negative values over land in the all-sky LW atmospheric absorption. Thus, the overall effect of the increased cloud in this region is to strengthen and spatially extend the atmospheric heating over ocean resulting from increased dust.

225   Over the Tropical Western Pacific Ocean we see a positive surface all-sky ERF (Figure 2d, S15), which is attributable to reductions in clouds (Figure 3, d) as opposed to changes in dust. In this region radiative effects due to changes in dust are negligible (Figures S5, S12) and therefore cloud effects dominate the all-sky ERFs (Figures S8, S15). At the surface, decreased cloud results in a small negative SW all-sky ERF, a positive LW all-sky ERF and as a consequence a positive total all-sky ERF (Figures S16, S17 and S15). At the TOA, reduced cloud results in a positive SW cloud ERF, a negative LW cloud ERF

230   and a total cloud ERF which is positive, albeit slightly patchy (Figures S25, S26, S24 respectively), indicating a dominance of the SW cloud ERF, whereby less SW radiation is scattered to upwards, resulting in a warming. The all-sky atmospheric heating is negative (fig 2c) which is also driven by cloud reductions causing a LW all-sky atmospheric cooling (Figures S23 and S21).

235

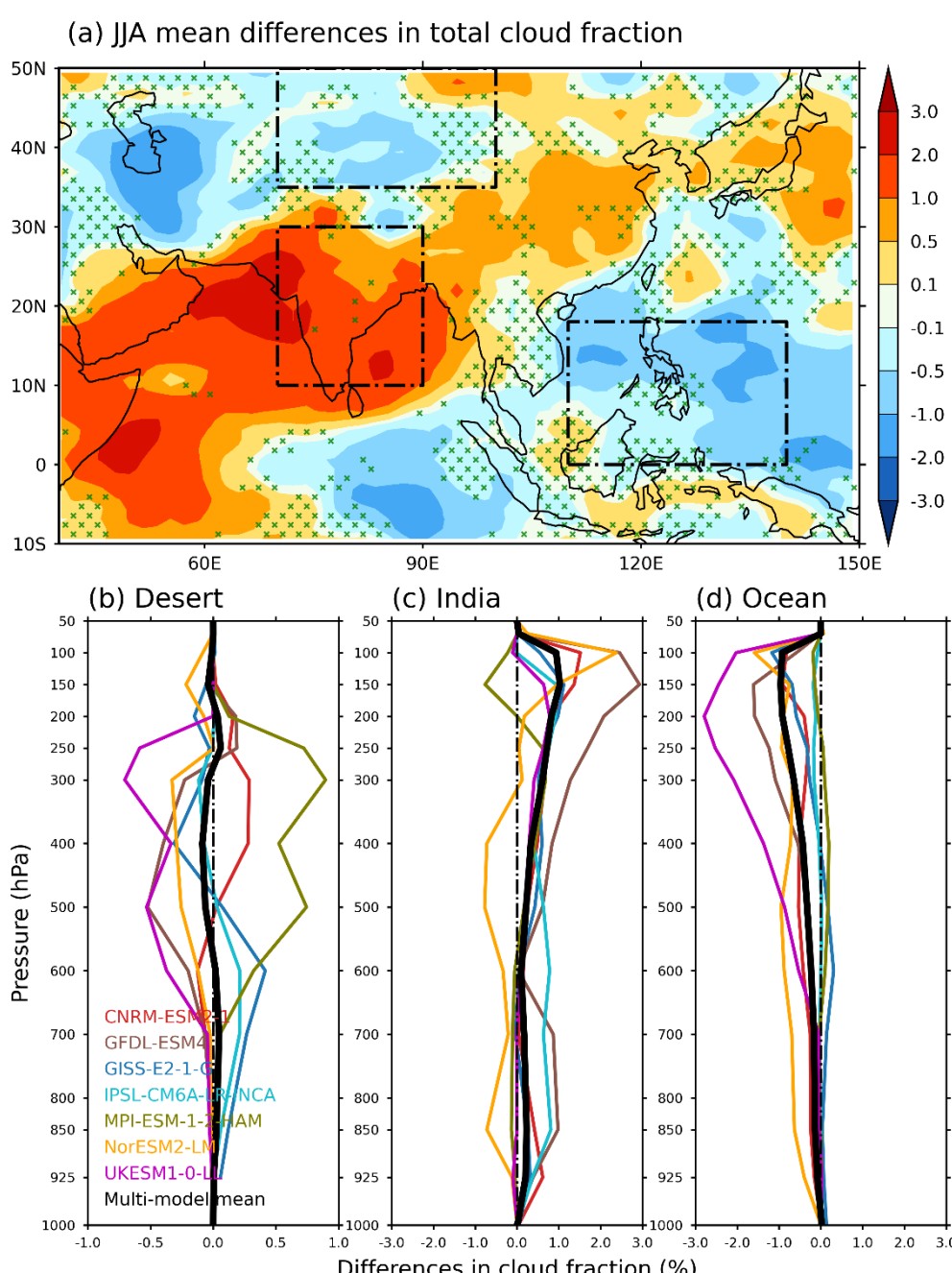

**Figure 3: JJA mean changes due to doubled dust emissions in (a) multi-model mean of total cloud fraction (%) and (b-d) vertical profiles of cloud fraction averaged within the three boxes in (a). Coloured curves represent individual models, and black the multi-model means. Green hatches in (a) denote where ≤4 models have the same sign as the multi-model mean.**

The dust-induced atmospheric absorption leads to a north-south alteration in energy distributions, as demonstrated by the changes in the Asian zonal mean atmospheric absorption (Figure 2e) and surface ERF (Figure 2f). The asymmetry is pronounced over the dustiest regions between 40E and 100E, encompassing the Arabian Peninsula, The Middle East, India and the Taklamakan desert, and is weaker over the East Asia-Western Pacific region. We show in Sections 3.3 that such changes have important fingerprints on dust-induced precipitation and circulation changes.

## 3.2 Temperature Response

Figure 4 shows JJA mean near-surface temperature changes in response to increased dust emissions. The temperature response is characterised by a cooling of the Indian subcontinent (-0.17 K) and the Chinese desert regions (-0.12 K) in the MMM (Figure 4h). The cooling is consistent across most models and is the largest in GFDL-ESM4 over India (up to -1.8 K), and in IPSL-CM6A-LR-INCA over the Chinese deserts (around -1.05 K). However, models disagree markedly with each other about the pattern, and even the sign, of the temperature responses over much of the rest of the domain (see locations of green hatching in Figure 4h). Over these regions, as opposed to the cooling seen in other models, the CNRM-ESM2-1 (Figure 4a), MPI-ESM-1-2-HAM (Figure 4e), and UKESM1-0-LL (Figure 4g) models simulate wide-spread warming. These models are also the ones with the lowest DOD climatology (Figure S1) and simulate the smallest DOD changes (Figure S4) there amongst the seven models. This uncertainty demonstrates the crucial importance of better observationally constrained representation of dust processes in climate models for simulating the dust-climate interactions.

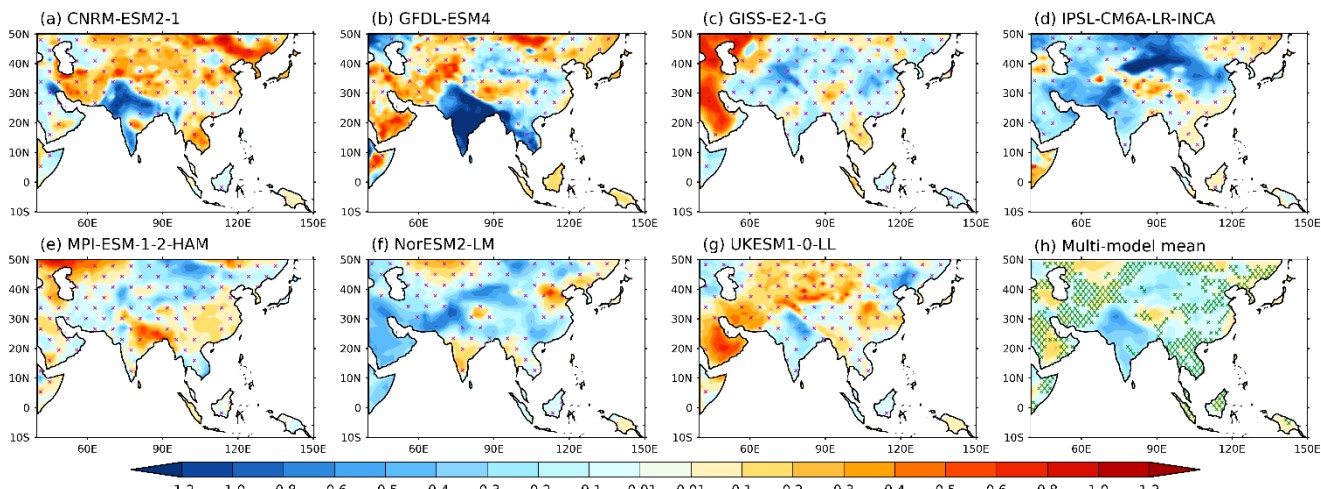

Figure 4: JJA mean changes in near-surface temperature due to doubled dust emissions in (a-f) individual models and (h) the multi-model mean. Purple hatches indicate lack of statistical significance at the 10% level. Green hatches in (h) denote where ≤4 models have the same sign as the multi-model mean.

The temperature responses in individual models do not follow the all-sky ERF at TOA (Figure 2b, S8), which shows opposite signs over some regions such as India. Similarly, near-surface temperature responses do not appear to show much relation to surface all-sky ERF patterns either (Figure 2d, S15). Overall, dust emissions result in a general surface cooling of the Asian continent in most models. However, there are significant diversities in model-simulated pattern and sign of temperature changes, despite the relatively consistent changes in cloud and radiation across models. Such diversity is only partly explained

by the diversity in the models' simulated dust climatologies. Meanwhile, we show below that such diversity in surface temperature response is also intertwined with changes in precipitation and monsoonal circulation.

### 3.3 Precipitation and circulation responses

In this section, we turn to JJA mean changes in precipitation due to dust emissions while attempting to understand the underlying mechanisms by examining changes in the ASM.


Figure 5 shows the spatial patterns, as well as the zonal mean profiles over the South Asia region (60-100°E, Figure 5i) and the East Asia-Western Pacific region (120-150°E, Figure 5j), of precipitation changes in response to dust emissions. We note that large uncertainties in models' simulated precipitation changes are expected due to challenges in simulating the ASM (Wilcox et al., 2020; Wang et al., 2021) in addition to the diversity of the dust climatologies. Nevertheless, the precipitation

responses exhibit certain common robust features. Particularly, the increased precipitation over the Indian subcontinent (up to 5%) and Southeast Asia (i.e., Indonesia and Papua New Guinea south of the Equator), as well as the drying (~10%) of the Western Pacific Ocean.

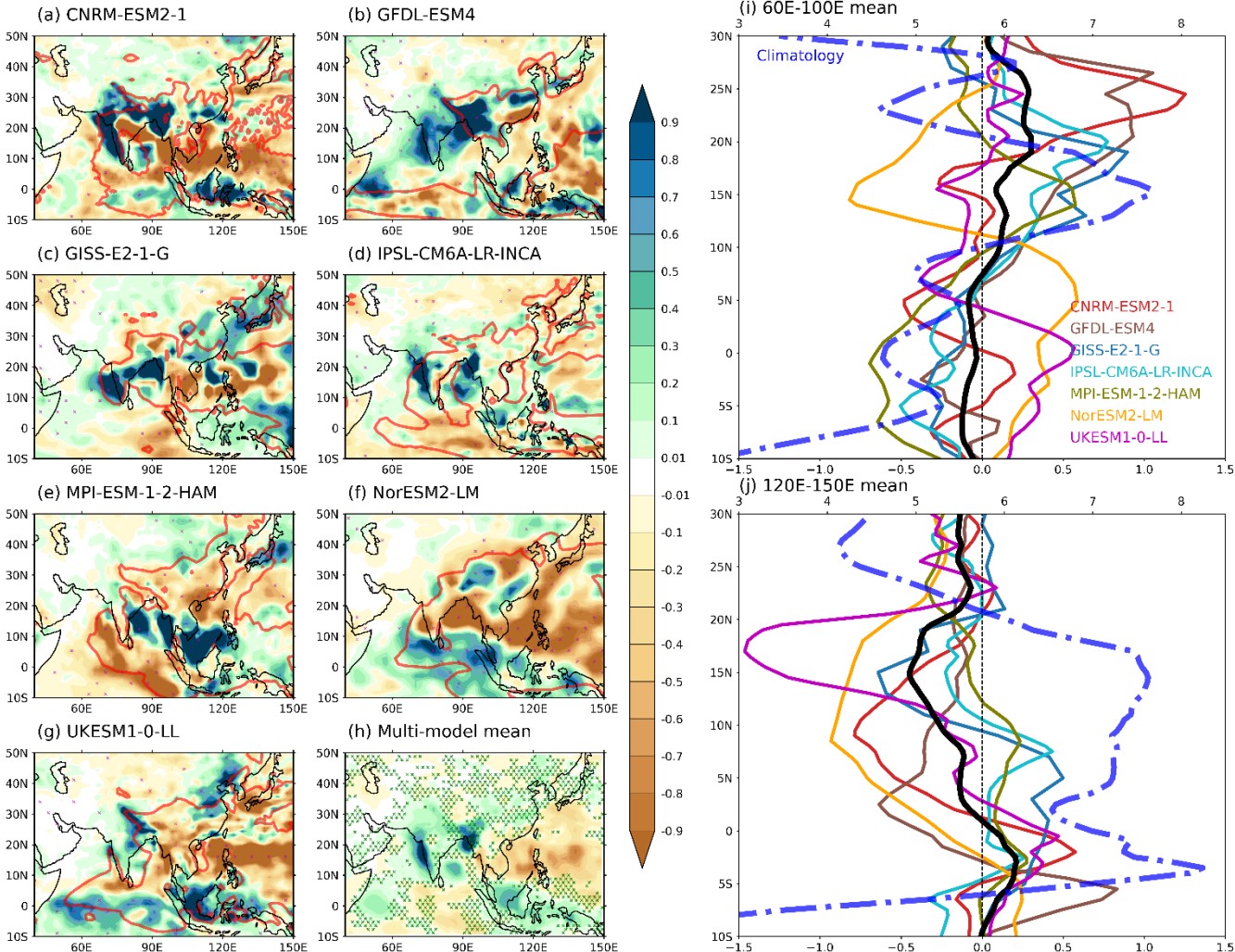

**Figure 5: JJA mean changes due to doubled dust emissions in precipitation (mm day⁻¹) in (a-f) individual models and (h) the multi-model mean. Red contours in (a-g) represent the 5 mm day⁻¹ JJA climatology derived from piClim-Control. Purple hatches indicate lack of statistical significance at the 10% level. Green hatches in (h) denote where ≤4 models have the same sign as the multi-model mean. Curves show the JJA zonal mean changes in precipitation (mm day⁻¹) averaged between (i) 60°E-100°E (South Asian region) and (j) 120°E-150°E (East Asia-Western Pacific). Coloured curves represent individual models (lower axis), black curves are multi-model means, and the blue dashed curves show JJA climatology (top axis) derived from the piClim-Control MMM.**

The MMM precipitation response (Figure 5h) is largely explained by changes in the vertically integrated moisture flux convergence (Figure 6a), whilst there is very little contribution from local convective processes, as demonstrated by the very limited changes in surface evaporation (Figure 6b). These, along with consistent changes in the 500-hPa vertical velocity (Figure 6c), demonstrate the predominant role of large-scale atmospheric circulation changes in shaping the fast precipitation response to dust emissions. The above is justified by careful comparisons of these fields (Figures S28-S29) to precipitation changes (Figure 5) in each individual model. For example, the pattern of precipitation increases over the Indian subcontinent

match well with the anomalous 500-hPa ascent and moisture convergence in most models. In comparison, the drying of the western Pacific Ocean is accompanied by strong anomalous descent at 500-hPa and moisture divergence in all models.

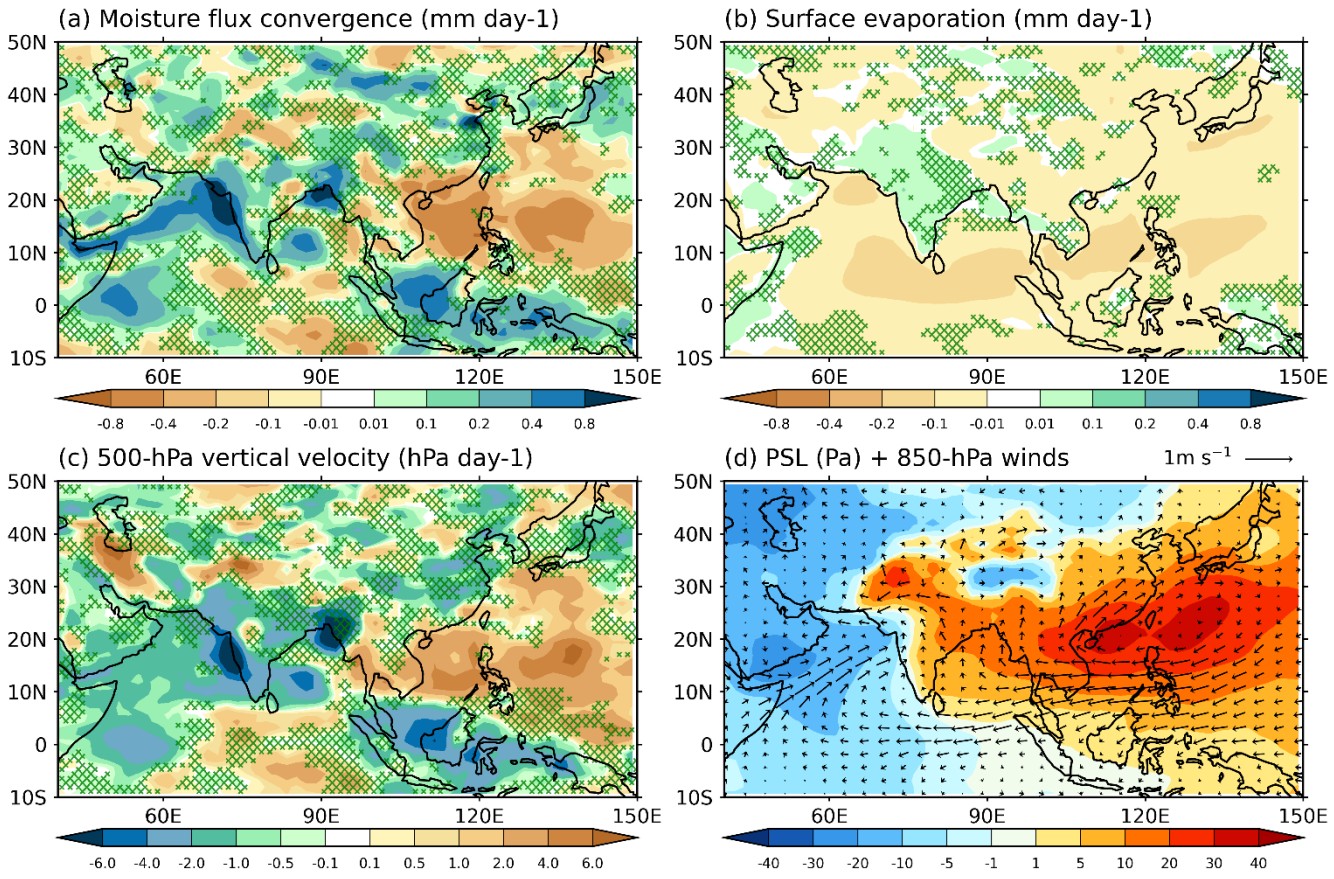


**Figure 6: JJA multi-model mean changes due to doubled dust emissions in (a) vertically integrated moisture flux convergence (mm day⁻¹), (b) surface evaporation (mm day⁻¹), (c) 500-hPa vertical velocity (hPa day⁻¹; negative values indicate increased upward motion), and (d) sea level pressure (colour; Pa) overlayed with 850-hPa winds (vector; m s⁻¹). Green hatches denote where ≤4 models have the same sign as the multi-model mean.**


The zonal mean precipitation changes show an enhancement of the ISM, with precipitation increasing over land and decreasing over the equatorial Indian Ocean in most models. That is, a northward shift of the rain belt over the ISM region (Figure 5, Figure 6). This is supported by changes in the 850-hPa winds in the MMM (Figure 6d) and in most models (Figure S30). Extensive lower tropospheric anti-cyclonic and south-westerly anomalies bring extra moisture from the Arabian Sea to the land. Over the Bay of Bengal, there are however anomalous southerlies that impede the climatological westerly flows; such southerlies are consistent with the pattern of enhanced precipitation there. The monsoonal precipitation increases lead to further cooling of the Indian subcontinent on top of the dust induced radiative surface cooling (Figure 4). The above changes in MMM

are also seen in most individual models. However, two models (MPI-ESM-1-2-HAM and NorESM2-LM) simulate weakened ISM circulation. This is consistent with the precipitation reduction over the Indian subcontinent (Figure 5e, f) and explains the
model-simulated warming there (Figure 4e, f) in these two models.

The importance of dust induced atmospheric absorption in changing monsoons and precipitation has been studied extensively (Maharana et al., 2019; Wang et al., 2020; Bercos-Hickey et al., 2020; Cruz et al., 2021; Jin et al., 2021; Balkanski et al., 2021) , with several different physical mechanisms proposed to explain their interactions. For example, snow darkening effects
(Sarangi et al., 2020) and the elevated heat pump (EHP) (Lau et al., 2006). Here we found that dust emissions cause enhanced atmospheric absorption over the Arabian Sea and South Asia (Figure 2) which is linked to enhanced moisture flux convergence via adjustments in circulations, and therefore an enhancement in the ISM in most models (Figure 6d). The enhanced ISM draws moisture from the oceans to the northern Indian subcontinent (Figure 6c), producing anomalous ascent and precipitation (Figure 5h), as well as collocated increases in high clouds (Figure 3). Although the increased dust absorption may also
contribute to cloud changes via the semi-direct effect (e.g. Doherty and Evan (2014)), the large-scale circulation changes here indicate that the South Asian cloud increases due to dust are circulation-driven. The total cloud atmospheric heating acts in the same direction as that of the dust (i.e. heating) and spatially extends the region of heating, further enhancing these effects. At the same time, strong southwesterlies within the monsoon are likely to transport more dust from the Arabian Peninsula to the Arabian Sea and Northern India. In doing so, the ISM is further enhanced through the EHP feedback loop brought about by
the enhanced upper-tropospheric meridional temperature gradient because of increases in dust absorption.

The East Asian Summer Monsoon (EASM) response to dust emissions is relatively weak and uncertain (Figure 6).. A westward extension of the West Pacific Subtropical High results in an enhanced monsoon flow over eastern China, and strong easterly anomalies over the tropical western Pacific Ocean and the South China Sea. The easterly anomalies disrupt the climatological
north-eastward transport of moisture flux from the oceans to the land (Figure S28). As a result, precipitation decreases over Southern China land areas, and only increases moderately over Northeast China in a few models (GFDL-ESM4, GISS-E2-1-G and UKESM1-0-LL) despite the enhanced monsoonal circulation over land. This demonstrates the large inter-model uncertainty in the response of the EASM to dust emissions in CMIP6 models that underpins the small response in the MMM. Such uncertainties can be attributed to several factors including model deficiencies in simulating the EASM (Wilcox et al.,
2015), the mixed circulation changes due to dust emissions, as well as the very low dust emissions over East Asia in most models.

The East Asia-Pacific region sees a southward shift of the Western Pacific ITCZ that is robust across models (Figure 5j). The southward shift of the Western Pacific ITCZ can be also seen in the spatial patterns of precipitation changes that feature a
north-south (drying centred around 15°N versus wettening centered around 5°S) dipole. The Western Pacific ITCZ shift is

consistent with the dust-emission-induced in surface radiative forcing (i.e.cooling in the Northern hemisphere) (Figure S15) due to atmospheric absorption (Figure 2c, f). This is consistent with (Evans et al., 2020) who found a linear relationship between dust emissions-induced hemispheric asymmetry in radiative forcing and tropical precipitation shift along global ITCZs. The southward shift of the Western Pacific ITCZ is accompanied by a general expansion of the Western Pacific Hadley

circulation and an enhancement of its ascending branch (not shown), as well as anomalous descent over the subtropical Western Pacific Ocean (Figure 6c, S29). The regions of anomalous descent are associated with collocated reductions in cloud fraction (Figure 3), anomalous surface high pressure (Figure 6d, S30), moisture divergence (Figure 6c, S28), and precipitation reduction (Figure 5). The equatorward limbs of the moisture divergence feed the Hadley circulation, forming a positive feedback loop between the drying of the subtropical Western Pacific Ocean and the southward shift of the Western Pacific ITCZ. The regions

of anomalous moisture divergence in some models also feed the tropical/subtropical anomalous easterlies that partly explain the mixed response of the EASM circulations.

Overall, we found a mixed response of the ASM to dust emissions which shows considerable diversity across models. The inter-model diversity in the atmospheric circulation response to dust is reflected in the uncertainties in models' simulated

temperature and precipitation changes. Nevertheless, the presence of a number of robust circulation changes across the models, and the fact that precipitation changes closely follow changes in circulation changes and moisture convergence, reveal the importance of large-scale atmospheric circulation changes in shaping temperature and precipitation responses induced by dust emissions. The impact of dust on the ASM suggests that deficiencies in ASM model simulations in general may be associated with the representation of dust processes. Meanwhile, the links between the shift of the Western Pacific ITCZ and dust may

have implications for the poorly constrained global ITCZs in most climate models (Samanta et al., 2019; Fiedler et al., 2020; Mamalakis et al., 2021). Specifically, since most models fail to capture the interannual to interdecadal variabilities of global and regional dust processes (Wu et al., 2019; Jin et al., 2021; Evan et al., 2014), they may also fail to reproduce the fingerprint of dust on the variability of global and regional ITCZs and monsoon systems on a number of timescales.

### 3.4 Relationship to Optical Properties

Here we investigate the relationship between the strength of SW absorption to the radiative and circulation changes. Ideally we would relate this back to the dust optical properties applied in each model (particularly the SSA). However, no information on the dust mass load or modelled size distribution (which evolves in space and time) is provided in these CMIP6 experiments

(and indeed in most CMIP6 AerChemMIP experiments (Zhao et al., 2022)). Despite this, we do have information on the visible wavelengths IRI (Table 1) for each model, which contributes to the absorption.

We calculated Pearson's correlation coefficients ($r^2$) and their significance (using a two-tailed t-test) for changes in DOD, atmospheric heating, and temperature over India (defined by the box shown in Figure 3a), and precipitation specifically for the Indian monsoon region (18-28N, and 75-85E). We selected this region since it demonstrated the strongest connection between changed dust and circulation changes. We also examined atmospheric heating normalized by changes in DOD, to account for the range of DOD changes across the models.

Interestingly, we find no relationship between the IRI and the change in clear-sky SW atmospheric heating due to doubled dust. However, we note that the model with the largest IRI (CNRM-ESM2-1) does give the largest change in normalized clear-sky SW atmospheric heating. We do, however, see a reasonably strong relationship between change in DOD and the change in SW clear-sky atmospheric heating across models ($r^2$=0.84, significant). Here GFDL-ESM4 shows the largest DOD change over India and also the largest atmospheric SW clear-sky heating, while CNRM-ESM2-1 had one of the smallest changes in both. The lack of relationship between IRI and clear-sky SW heating, in contrast to the strong dependence on the DOD change, points to the importance of simulated dust load in influencing atmospheric circulation. It also emphasizes the unavailable dust mass/size data in contributing to changes in both the total absorption and the DOD alongside the IRI. It appears that the change in dust burden inferred through the DOD, rather than the SW optical properties of the dust, is the dominant driver of changes in SW clear-sky absorption here.

The clear relationship between DOD change and atmospheric heating persists from the SW clear-sky, to SW all-sky absorption ($r^2$=0.81, significant), and also to total (i.e. SW plus LW) clear-sky ($r^2$=0.86, not significant) and all-sky ($r^2$=0.88, significant) absorption, due to the dominance of the SW radiative effect of the dust over the LW, as seen in Section 3.1. Additionally, models with a large change in SW clear-sky atmospheric heating produced greater ISM precipitation change ($r^2$=0.81, significant) and greater decreases in surface temperature ($r^2$ = -0.87, significant). Thus, the discrepancy in DOD change across models appears to explain the range of change in SW atmospheric heating under the doubled dust scenario, which goes on to cause the range of responses in precipitation and surface temperature. Since the change in DOD relates directly to the underlying model DOD climatology magnitude in each model, this suggests that the range of dust-induced circulation responses depend on each model's underlying dust climatology, which are hugely variable (Zhao et al., 2022).

In the absence of transported dust size data, we also compared the relationship between change in DOD and SW clear-sky atmospheric absorption to the maximum size of dust represented by the models' dust schemes (Table 1), since larger size contributes to greater SW absorption (Ryder et al., 2019). Again, no relationship was evident relating these variables to the maximum dust size, though this is perhaps unsurprising given that models have a tendency not to transport coarser dust particles far in the atmosphere, even if larger model size bins do exist in the dust scheme (e.g. Ratcliffe et al. (2024). Further,

it is unclear how well modal schemes may represent the complexities of the coarser end of the dust size distribution during transport (e.g. Jones et al. (2022)).

We note that the small number of models is not ideal for these statistical tests, and neither is the cluster of IRI values around small values (0.001-0.002i) with only one model with a much larger value (0.008i). We did not perform an analysis of the relationship between LW optical properties and dust ERF, despite their radiative importance, due to the LW optical properties being even more difficult to identify for each model than those of the SW spectrum, and the importance of other measures of dust such as plume altitude and size. We found that there was no relationship between change in DOD and clear-sky LW atmospheric heating.

## 4 Conclusion and Discussion

We investigated the fast ASM response to a doubling of global dust emissions in seven CMIP6 models. Our results offer a parallel to the impacts of preindustrial to present-day global dust emission changes since global dust emissions have approximately doubled since preindustrial times, as well as an insight into the concealed effect of dust on climate within the latest generation of climate models. We found that doubled dust emissions cause significant atmospheric absorption over Asia. This results in circulation changes: an intensification of the ISM (precipitation increases of up to 5%) exhibited by increased cloud and precipitation in this region, whereby the radiative effects of the increased cloud amplify the radiative effects from doubled dust, further enhancing circulation changes, despite a surface  cooling of the Indian subcontinent due to increased precipitation. Additionally, we find a southward shift of the Western Pacific ITCZ as a result of the circulation changes from dust absorption. These demonstrate important fingerprints of dust emissions on the ASM through dust absorption-induced large-scale circulation changes. For the ISM, we find that the strength of the monsoon response depends on the magnitude of the change in dust shortwave absorption, which is related to the change in DOD and therefore the underlying model dust climatology, which is hugely variable across models. We found no relationship between dust-driven atmospheric absorption and dust imaginary refractive index, with DOD changes being the primary driver. However, lack of dust size and mass data in the CMIP6 experiments prevents a full analysis of the relationship between dust optical properties and atmospheric absorption effects. There are also considerable uncertainties in models' simulated dust processes and in the large-scale circulation changes in response to dust emissions across models. Particularly, the model climatology of dust emission and loading seems to play a role in model-simulated climate responses. This demonstrates the importance of observationally constrained dust processes and properties, particularly absorption and DOD, for constraining the ASM, and better constrained large-scale circulations for more reliable simulations of dust-climate interactions.

We provide the caveat that the responses to dust emissions might be incomplete in the model simulations we analysed here (Zanis et al., 2020). Firstly, the CMIP6 models poorly capture and underestimate dust load over the Indian subcontinent (Zhao et al., 2022). Therefore, the dust induced atmospheric absorption there and its impacts might also be underrepresented. Secondly, the contribution of Asian dust emissions to the global total is found to be underestimated by present generation

climate models (Kok et al., 2021a). Thirdly, the significant low biases in the size and size distributions of dust particles in present generation climate models (Ryder et al., 2019; Adebiyi et al., 2023a; Huang et al., 2021) may also mean underestimated atmospheric absorption and reduced longwave dust-radiation interactions, which could alter the impacts of doubling dust emissions. Although recent results (Di Biagio et al., 2019) suggest that climate models tend to apply values of the imaginary part of the refractive index of dust which are too high in the shortwave spectrum, we find that most models examined here

apply values within a reasonable range. Modelled size-resolved mass concentration data is generally not available for the CMIP6 experiments. Inclusion of such data in future CMIP experiments would be beneficial for understanding the breadth of interactions from dust optical properties through to climate and circulation, and is recommended for inclusion in future experiments. We also urge the modelling community to make model dust optical properties, in both the shortwave and longwave spectrum, more easily available and up to date.Assumptions and uncertainties around these parameters in climate

models will have great implications for model-simulated signs and magnitudes of the climate responses to dust emissions. Finally, the experiments analysed here are atmosphere-only simulations. The pattern and magnitude of the response to dust is likely different in fully-coupled climate models, as has been demonstrated in several studies of the response to anthropogenic aerosols (Ganguly et al., 2012; Samset et al., 2016; Voigt et al., 2017), since the anticipated cooling effect of dust on sea surface temperatures may have impacts on monsoon circulation.


We acknowledge that the climate response to dust emissions are still highly uncertain in climate models, given the large diversity reported here. However, whether conclusions drawn from the seven models analysed here are just a reflection of a sample of many more CMIP6 models is unknown, and we note that the number of models participating in this experiment is fairly low (seven). For example, we report model agreement where the number of models in agreement with the MMM is 5

out of 7 – having better statistics and model participation is desirable. This warrants a community effort to better understand and simulate dust processes in climate models given their potential significance in accurately simulating other intertwined processes. The responses presented here are due to global dust emissions, and we recommend further model experiments to compare the impacts of local vs. remote dust emissions. Dust as ice nuclei and related processes are still missing in most models, which may affect model-simulated dust-climate interactions (Froyd et al., 2022). We noted however that two models

(MPI-ESM-1-2-HAM and NorESM2-LM) have parameterised dust particles as ice nuclei. Nonetheless, changes in ice-cloud microphysics (Figure S31) in these two models are insignificant, high cloud changes in dusty regions are of opposite signs, such that the inclusion of dust-IN interactions do not explain their differences compared to other models. We therefore suggest further studies to understand the possible reasons behind this.

In summary, we found that doubling global dust emissions results in enhanced atmospheric absorption over the Arabian Sea and South Asia, causing an intensification of the ISM resulting in increased precipitation over the Indian subcontinent and a subsequent surface cooling, a mixed response of the EASM, and a southward shift of the Western Pacific ITCZ in the CMIP6 models. These responses feature large inter-model diversities that are intertwined with diversities in model-simulated large-scale circulation changes, although the magnitude of these changes depends on the magnitude of the dust-induced atmospheric

absorption, strongly related to the dust optical depth. These responses may only represent a certain fraction of the full response. it is therefore possible that dust may play an even greater role in global climate interactions than we present here. More importantly, we suggest that accurate representation of dust should be a consideration in efforts to reduce monsoon biases in climate models, and dust may represent an important feedback in future projections of both the ASM and the regional and global ITCZs.

**Data Availability**


This work uses simulations from the Coupled Model Intercomparison Project (Phase 6; https://www.wcrp-climate.org/wgcm-cmip, World Climate Research Program, 2020). Model outputs are available on the Earth System Grid Federation (ESGF) website (https://esgf-data.dkrz.de/search/cmip6-dkrz/, Earth System Grid Federation, 2020).

**Author Contribution**

CR and LW designed the experiment; AZ carried out data processing and analysis with input from CR and LW; AZ and CR wrote the manuscript with contributions from LW.

**Competing Interests**

Laura Wilcox is a member of the editorial board.

**Acknowledgements**

This work and its contributors AZ, CLR, and LJW were supported by the UK-China Research and Innovation Partnership Fund through the Met Office Climate Science for Service Partnership (CSSP) China, DAHLIA project, as part of the Newton Fund. CLR was supported by a UK Natural Environment Research Council (NERC) independent research fellowship (grant no. NE/M018288/1). We acknowledge the World Climate Research Programme, which, through its Working Group on Coupled Modelling, coordinated and promoted CMIP6. We thank the climate modelling groups for producing and making

 available their model output, the Earth System Grid Federation (ESGF) for archiving the data and providing access, and the multiple funding agencies who support CMIP6 and ESGF.

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
