# Peer review of "The key role of atmospheric absorption in the Asian Summer Monsoon response to dust emissions in CMIP6 models"

_EGUsphere, 2023_

## Author Comment (AC1)

**Response to Reviewers**

**Reviewer 1**

The manuscript leverages on existing CMIP6 simulations, in particular doubled dust experiments within AerChemMIP, to infer model impacts of dust on the Asian Summer Monsoon, using 7 pairs of simulations with different models in atmospheric only configuration. Relations among simulated variables from the multi-model mean and (in most cases) individual models are shown and discussed to provide a coherent picture of possible underlying mechanisms of observed dust perturbations to the system. I found the manuscript interesting and generally well written. I do have, nonetheless, a couple of general remarks and a few specific comments.

Starting from the title there is some emphasis on "atmospheric absorption". However, if space is dedicated to the consequences, I found little details on its causes, both in terms of discussing the possible mechanisms, and in terms of displayed variables. In particular, it is not very clear which modeled processes are being actually simulated (SW-LW interactions, CCN/IN) and not enough details are provided in terms of shown variables (e.g. only net ERF is shown for individual models; dust-cloud interactions are deemed insignificant but not shown). I would suggest discussing a bit more in detail possible mechanisms at least in the introduction, and be more thorough in reporting the relevant available variables.

We thank the reviewer for their positive and insightful comments. We have addressed the points raised and these are detailed below.

We have extended and improved Table 1 (model information), which now clearly states which model processes are included (all models include SW and LW dust radiative interactions, IN/CCN interactions with dust are stated), and a full range of ERF variables is now provided in the supplement (TOA/surface/atmospheric heating; all-sky/clear-sky/cloud; SW/LW/Total).

There is also, here and there, some lack of attention for details (plots in the supplement slightly different than the corresponding versions in the main text, missing references, imprecision in reporting models' relevant features, ...) that warrants an accurate revision.

Please see responses to specific comments for full details. Broadly, we have tightened up and improved model information (Table 1) and revised and extended the figures in the Supplement.

Specific comments

38 > What do you mean by "most of these mechanisms are model-based"? Please rephrase

Agreed, we deleted these words.

86-88 > This sentence seems a bit contradictory: if emissions data files could be scaled, that could imply that dust emissions are prescribed. So why do you state that all models calculate dust emissions online? Please be more specific about the different strategies of the simulations involved in this study, if known, otherwise comment on that.

All the models use online emissions, and double these in the doubled dust experiments. We deleted the reference to the emission files as it was not helpful. (This may have been relevant if models prescribing dust emissions had participated in the experiment).

91-92 & Table 1> In this context, we are not really interested in whether dust-cloud interactions could be turned on in a model, but rather if they were actually activated in these simulations. By the way, I would strongly suggest reviewing all the information reported in the table. For instance, the IPSL model version used in CMIP6 does not have dust-cloud interactions of any kind, and the reference provided describes a post-CMIP6 version of the model, with 4 dust modes – not one. There may be analogous imprecision concerning other models. Please double-check everything carefully.

This was the intention – to describe whether or not each model utilizes the interactions in the experiments presented. The text has been clarified on this point.

We have extensively checked, revised and expanded on the detail in Table 1, correcting information where necessary.

98 > What kind of interpolation did you use? Bi-linear?

We used a first order conservative interpolation for fields that request integral of the source field across the regridding – dust emissions, for example. For all other variables, we used bilinear interpolation using the Earth System Modeling Framework (ESMF) framework. This information has been added to the Methods section.

109-110 > "regions where there are inconsistent responses across models, defined as regions where ≤4 of the 7 models have the same sign as the MMM". This definition is clearly stated, and in most cases individual model results are available in the supplement and differences discussed in the main text. However, often in the text expressions like "most models" (e.g. line 172) indicate a 50% + 1 situation, rather than a clear majority. This highlights the inherent difficulties of this exercise. Maybe add some comments about this in the text.

We added the following line to the conclusions, "and we note that the number of models participating in this experiment is fairly low (seven). For example, we report model agreement where the number of models in agreement with the MMM is 5 out of 7 – having better statistics and model participation would be desirable."

126 > "hatches in (f) denote statistical insignificance at the 10% level". This phrasing is a bit confusing, considering what we see in the plot. Please adjust/rephrase. In addition, here and everywhere else in the captions of figures in the main text and supplement, clarify the metric used, e.g. using a two-tailed student's t test ...

The caption on the figure has been corrected.

We have clarified the use of the specific significance test in the methodology section.

133-134 > The MMM represented in both Figure 2b and Figure S6h shows differences at least in the hatching. If a different criterion and/or subsets of simulations were used in the submitted version of the manuscript, compared to what was initially tested for this work, please revise and make sure figures are consistent - this applies to many of the plots. In fact (minor) differences in plotted variables values (considering obvious cases where color scales match) and/or hatching also appear in Fig. 2a vs S5h, Fig. 2c vs S7h, Fig. 6a vs Fig. S11h, Fig. 6c vs S12h.

The reason for these differences was that different densities of hatches were used for visual purposes. All figures have now been reproduced so that each hatch represents a single model gridcell and are consistent between the main article and supplement.

135-136 > I would add something like "although not necessarily statistically significant"

Done

147 > It would be interesting to see also the individual models SW vs LW partitioning, as done for all other variables.

We have added the following figures with additional variables for individual models and the MMM to the supplement: all-sky ERF TOA/surface/atmospheric heating, clear-sky ERF TOA/surface/atmospheric heating, shown as SW, LW and total.

150 > The notation "Ocean" is rather misleading, considering that the Arabian Sea and portions of the Indian Ocean show the opposite situation. I would suggest using again "Tropical Western Pacific Ocean".

We have changed the wording here to clarify.

152-155 > "robust common patterns" seems too strong a statement, compared to what we actually see. Also "All models show that such changes come from … changes in mid-level clouds (700-200 hPa) above the Chinese deserts (Figure 3b)" is inconsistent with at least two models showing the opposite. Please re-write this paragraph accounting for the actual variability emerging from the figures. More in general, try to be more precise when discussing the variability among individual models, and otherwise specify when you are discussing the ensemble mean specifically. This is done in some passages, but not systematically.

We removed the word 'robust' here and added, '…generally showing increased cloud fraction in these regions…' We also added text to more accurately reflect the data shown in Figure 3b-d.

156-157 > All statements about dust-cloud interactions seem highly speculative, unless we see specific diagnostics (for the subset of models actually parameterizing this process in the simulations at hand). There is also the potential contribution of semi-direct effects, which is not mentioned.

On the point of dust-cloud interactions, please see our response to point further on.

On the topic of the semi-direct effect, this is represented in the model simulations through the dust-radiation interactions and their contribution to atmospheric heating. It can be considered somewhat unclear in terms of where to draw the line in terms of defining a semi-direct effect. In the experiments we present, atmospheric heating alters moisture flux convergence, and therefore circulation and cloud, i.e. circulation-driven. The semi-direct effect is often defined as a more local impact of aerosol heating on cloud formation or dispersion. We added the following sentence to section 3.3, "Although the increased dust absorption may also contribute to cloud development via the semi-direct effect (e.g. Koren et al., 2008), the large-scale circulation changes here indicate that the South Asian cloud increases due to dust are circulation-driven."

163 > It is not clear what you mean by "hemispheric asymmetry". Since you were just discussing the Indian subcontinent (and later on you add the Arabian Peninsula etc.) versus the "Tropical Western Pacific Ocean", one would imagine that you are implying a zonal asymmetry. Please clarify.

We have added the words, 'north-south alteration' and deleted 'hemispheric asymmetry.'

169 > "in response to INCREASED dust emissions"

Added

173-175 > I would leave out the Arabian Peninsula, as it does not fit the given description of what happens in the listed models. It is also not clear to which regions in particular the definitions of Central and East Asia apply. In fact, these two sentences are overall quite unclear / imprecise in describing the corresponding plots. Please clarify.

We changed the text to 'over much of the rest of the domain' and reference the reader to the green hatches in Figure 4h.

184-186 > This may hold for India, but not for instance for the Arabian Peninsula. Hard to generalize.

We deleted these sentences and slightly reworded the earlier one, since we show later that the precipitation changes explain the surface temperature changes.

210-211 > Is climatology from the MMM?

Yes, climatology is from the MMM. This is now included in the caption.

238-239 > (Maharana et al., 2019; Bercos-Hickey et al., 2020; Cruz et al., 2021; Lau et al., 2006). These references are not in the bibliography. (Balkanski et al., 2021) could be a relevant reference in terms of mechanism, however it is not pertinent considering the way the paragraph is currently phrased, i.e. it does not focus on the ISM, but on the West African Monsoon; please rephrase.

The references have been added to the bibliography and the sentence adjusted.

262-263 > I fail to clearly see this hemispheric asymmetry from the results presented in this study. (Evans, 2020) is also missing from the reference list

We do not show hemispheric figures since we focus on the ASM region. We have adjusted the wording slightly to reflect this.

Evans et al. (2020) has been added to the references.

289-290 > Same as above

The text has been modified away from the word 'hemispheric' – similar for the abstract and other occurrences.

330-331 > It may be worth showing

This comment refers to the statement, "We noted however that two models (MPI-ESM-1-2-HAM and NorESM2-LM) have parameterised dust particles as ice nuclei. Nonetheless, changes in ice-cloud microphysics (not shown) in these two models are insignificant and do not explain their differences compared to other models."

Unfortunately here we are limited by the available diagnostics provided by different modelling centres. For NorESM-LM (but not MPI-ESM-1-2-HAM), cloud condensed ice diagnostics are available, shown below (panel a). Hatches indicate where the changes in cloud condensed ice are significant, and are limited to areas over the Indian Ocean and northeast Asia.

In the figures below, NorESM has increased condensed ice and liquid water over the Indian Ocean and Tibetan Plateau. However this only partially matches up with locations of increased cloud fraction for this model (Tibetan Plateau and the Middle East peninsula, S27).

This figure is now provided in the Supplement (S31) and cross-referenced in the text.

[Figure]

Figure: NorESM2-LM changes in cloud properties due to doubled dust emissions

**Review 2**

Summary

This study investigates the response of the Asian Summer Monsoon (ASM) to a doubling of dust emission in a subset of Coupled Model Intercomparison Project Phase 6 (CMIP6) models. The manuscript describes differences in dust effective forcing (net flux differences between 1x and 2x dust emission) and the resultant changes summertime atmospheric moisture and regional circulation. The main findings of this study are that the doubling of dust results in atmospheric heating over the continent that, via the resultant hemispheric energy imbalance and circulation response to that imbalance, enhances the monsoon over the Indian subcontinent. There was no strong model consensus regarding the effect of the doubling of dust emission on the East Asian Monsoon (which I suggest could imply that there is not a strong one).

The topic of how dust emission affects monsoonal precipitation is interesting and the CMIP6 data set is a novel one to use for the analysis. My critique of this manuscript is that it mainly describes discrepancies in the models' response to the dust radiative forcing, rather than

providing any insight into the causes of those discrepancies, where the latter would make for a more interesting and potentially useful study. The authors imply that it is not possible to understand the source of the model discrepancies because the requisite data is not available (Lines 313-315). However, I think a little digging into the models' representation of dust could at the very least tell us something about the prescribed dust optical properties, which the authors indicate as being fundamental to the forced response. The authors could even diagnose which models have larger dust absorptivity by examining the differences (differences between the 1x and 2x dust emission) in SW atmospheric net fluxes. After reading the manuscript I'm left with the impression that the authors feel like it's just not worth trying to understand the sources of the model discrepancies.

We thank the reviewer for their positive and insightful comments. We have significantly expanded on the discussion around the absorption across models, and how this relates to optical parameters set by each model. Further details are below.

Specific Comments:

1. Abstract: "Our results demonstrate the central role of dust absorption in influencing the ASM". I understand that this is implied by the analysis. However, isn't it possible to identify which models exhibit the strongest (presumably) solar absorption by dust and then demonstrate that the magnitude of the effects on the ASM are somehow correlated to that absorptivity?
   Please also see response to point #9.
   Thank you for this useful suggestion. We have expanded on this point and now show that the absorptivity relates to the change in DOD, and that this goes onto correlate to the change in precipitation and surface temperature over India. Interestingly the clear-sky solar absorptivity is unrelated to the imaginary refractive index, but is related to DOD change. We now discuss this in the manuscript, including the influence of the (unavailable) size distribution/mass data.

2. Line 20: I don't think the Kok 2018 reference is appropriate here.
   Removed

3. Line 34: Extra ")"
   This is correct (brackets around both references and i.e. statement) and therefore unchanged.

4. Line 50: "found" is redundant.
   Sentence reworded

5. Line 136: Should be Fig 2b
   Changed

6. Line 145: Should this be "a doubling of dust emission"?
   We refer the reviewer to our line in the Methodology, "Note all the changes presented here are due to a doubling of global dust emissions, and we refer to this as 'dust emissions' for simplicity," which we believe covers this point.

7. 2a & b: Why is there more inter-model agreement in the all-sky ERF than in the clear-sky ERF over the Indian subcontinent? If the model differences in clouds are forced by the changes in the dust ERF, I'd expect more uncertainty in the all-sky ERF than in the clear-sky.
   This is because at the TOA the clear-sky ERF can be positive or negative over India due to differences in how models represent dust, in terms of how the strengths of the SW (mostly negative) or LW (positive) dust radiative effects cancel out. Once cloud is

included (all-sky), the TOA ERF is shifted towards positive values, such that most models agree on an overall positive all-sky ERF.
This is now discussed in Section 3.1.

8. Line 164: The hemispheric asymmetry in dust radiative forcing has been documented and examined by other studies, and I think it would be reasonable to cite those here.
We reworded this section based on the comment of reviewer 1.
We cite various studies examining hemispheric asymmetry in section 3.3 where we discuss the circulation response.

9. Lines 172--178: throughout this manuscript, I think the authors are missing an opportunity to provide insights into the causes of the discrepancies among the modeling, and here is one such example. The authors describe the differences in the modeled responses to the doubling of emission over Asia land surfaces. They note that four models show warning temperatures, while the others show cooling. The authors go on to point out that these models' mean states have the least amount of emission, and consequently the smallest increase in dust optical depth for the experiment. My suggestion is that the authors propose and test a hypothesis explaining why there may be a connection between the apparent over-land warming and small change in the absolute dust emission. This would be far more interesting and insightful than just describing the differences between models.

We thank the reviewer for this comment, and have added a new section (3.4) to the results section to add more to the study from this perspective.

We calculated correlations between various variables in order to provide a more useful link between the dust information available for each model (including the newly-added SW refractive indices), and the radiative/circulation responses

In summary, we find that there is a strong, statistically significant relationship between Indian change in DOD and clear-sky SW atmospheric absorption, but no relationship between the SW imaginary refractive index (IRI) and clear-sky SW atmospheric absorption. There is also a strong correlation between clear-sky SW atmospheric absorption and surface temperature and ISM precipitation. Since the change in DOD relates to each model's underlying DOD and emission climatology, this variability appears to be the cause of the range in circulation and precipitation responses seen across the models in the doubled dust experiment.

We have also expanded on the radiative description analysis and discussion in section 3.1-3.2 to further pin down the strong contribution from the dust SW absorption.

10. Lines 184—186: Please be clear if you are discussing the clear-sky or all-sky ERF. It would be nice to not have to reference the figures to obtain this information.
Done here, and throughout the manuscript.

11. Line 186: I don't understand what is meant by "the central role of dust-radiation interactions in changing the surface radiation budget and temperature". Are the authors suggesting that the surface energy budget over Asia is, to first order, controlled by dust? I doubt this is the case since it's a rather extreme claim, and suggest that the text be revised to be more precise/clear.
Changed to read, "these emphasize the importance of dust-radiation interactions..."

12. Line 249: The statement "The East Asian Summer Monsoon (EASM) presents a mixed response (Figure 6d) to dust emissions" is a little strange to me since it implies that the models are telling us something concrete about the effect of dust radiative forcing on the monsoons, yet in the paper the authors are telling us that the models are all over the place and so we shouldn't trust them.

Here we describe how the models respond differently in the EASM behaviour with respect to the increase in dust emissions. In this case the impact of the changed radiative forcing due to dust changes, and hence impact on circulation wrt the EASM is not robust across models. The sentence now reads, "The East Asian Summer Monsoon (EASM) response to dust emissions is relatively weak and uncertain."

13. Line 314: The authors state that modeling centers should provide more information on dust single scatter properties and size distributions. However, I'm sure that these data are, to some extent, available in the mode descriptions. I wonder if the authors could do a more thorough job of diagnosing the underlying causes of inter-model discrepancies by comparing, for example, differences in dust single scatter albedo, since this would play a very large role in determining net radiative forcing.

Table 1, providing model information, is now significantly expanded.

We have spent a lot of time searching through model descriptions and leveraging contacts to determine information about dust optical properties. Much of this information is buried very deep in older publications, sometimes out of date, and occasionally not provided at all. Nevertheless, we have identified the dust imaginary refractive index (IRI, which affects the absorption) applied in each model at 550 nm and listed this in table 1.

However, it is not simply the IRI which determines the effect of dust on atmospheric absorption. In addition to the dust loading, we need to know the single scattering albedo (SSA) of the dust, which is determined by shape (typically spherical in models), imaginary refractive index (IRI) and additionally the particle size distribution (PSD) of dust simulated by the models. PSD will vary with model dust emission scheme, model climate (e.g. surface winds and moisture), model dust transport and deposition processes, and will evolve with transport. Models typically use look up tables to calculate optical properties such as SSA with the evolved PSD, which varies with space and time. CMIP6 models did not report SSA diagnostics or size-resolved mass diagnostics, so we cannot (for example) calculate SSA or AAOD, for example as implemented by Adebiyi et al. (2023). Model descriptions in publications generally do not report SSA information since it varies with the simulated dust size distribution. Therefore we do not have the full information to diagnose inter-model differences in SSA and the impact on dust atmospheric absorption.

Nevertheless, we now attempt to relate model dust IRI to clear-sky dust absorption. Please see above response to point #9 regarding this analysis.

14. I was surprised that the authors didn't examine the cloud radiative effect in the experiments, since changes in cloudiness play such a strong role in the response. This could be a useful metric.

We now include cloud ERF at the TOA in the supplement (figures S24-26) and discuss this in the article.

15. With regards to diagnosing temperature changes, I suspect that an energy budget analysis would allow the authors to make a stronger and quantitative case for the relevant factors shaping the differences in temperatures between the experiments.
We have further analysed and provided plots of separated SW/LW/total clear-sky and all-sky ERFs at the TOA, surface and atmosphere. Although a surface energy budget analysis would certainly be interesting, unfortunately resources do not permit us to investigate this further.